

# Consistency and representativeness of integrated water vapour from ground-based GPS observations and ERA-Interim reanalysis

Olivier Bock[1] and Ana C. Parracho[2]

[1]IPGP, IGN, ENSG, Universite Paris Diderot, Sorbonne Paris Cite, UMR 7154 CNRS, Paris, France
[2]LATMOS-IPSL, CNRS UMR8190, Sorbonne Université, Paris, France

*Correspondence to:* Olivier Bock (Olivier.bock@ign.fr)

**Abstract.** This study examines the consistency and representativeness differences of daily IWV data from ERA-Interim reanalysis and GPS observations at 120 global sites over a 16-year period (1995-2010). Various comparison statistics are analysed as a function of geographic, topographic, and climatic features. A small ($\pm 1$ kg m$^{-2}$) bias is found in the reanalysis
across latitudes (moist in northern and southern mid-latitudes and dry in the tropics). The standard deviation of daily IWV differences is generally below 2 kg m$^{-2}$ but peaks in the northern and southern storm-tracks regions. In general, the larger IWV differences are explained by increased representativeness errors, when GPS observations capture some small-scale variability that is not resolved by the reanalysis. A representativeness error statistic is proposed which measures the spatiotemporal variability in the vicinity of the GPS sites, based on reanalysis data at the four surrounding grid points. It allows to predict the
standard deviation of daily IWV differences with a correlation of 0.73. In general, representativeness differences can be reduced by temporal averaging and spatial interpolation from the four surrounding grid points. A small number of outlying cases (15 sites) which don't follow the general tendencies are further examined. It is found that their special topographic and climatic features strongly enhance the representativeness errors (e.g. steep topography and coast-lines, strong seasonal cycle in monsoon regions). Discarding these sites significantly improves the global ERA-Interim and GPS comparison results. The
selection of site *a priori*, based on the representativeness error statistic, is able to detect 11 out of the 15 sites and improve the comparison results by 20 to 30%.

## 1. Introduction

Quantifying the global atmospheric moisture distribution and its variability across time scales remains a challenge to the climate community. Atmospheric reanalyses offer a comprehensive representation of the various components of the
hydrological cycle, among which precipitation and evaporation are the dominant terms at the larger space and time scales. However, both quantities result from model integrations and are not strongly constrained by observations (Trenberth et al., 2011). The difference of precipitation minus evaporation corresponds to the net vertically integrated atmospheric moisture convergence, a quantity which can also be computed from analysed three-dimensional moisture and wind fields which benefit directly from the assimilation of observations (Trenberth and Fasullo, 2013). However, due to the high spatiotemporal



variability of atmospheric moisture, the quality of moisture fields in the reanalyses remains limited, especially in data-sparse areas (Trenberth et al., 2005; Meynadier et al., 2010).

Ground-based Global Positioning System (GPS) Integrated Water Vapour (IWV) observations have been used for some time as an independent validation source for global atmospheric reanalyses over limited regions and periods (Hagemann et al., 2003; Bock et al., 2005; Heise et al., 2009; Bock and Nuret, 2009; Bock et al., 2016) and moist atmospheric process studies (Bastin et al., 2007; Bock et al., 2008; Koulali Idrissi et al., 2012; Means, 2013; Adler et al., 2015; Khodayar et al., 2018). More recently, the value of continuous long time series of GPS IWV data has been investigated for the purpose of studying global and regional climate variability and validating climate models (Nilsson and Elgered, 2008; Vey et al., 2009; Roman et al., 2012; Ning et al., 2013; Chen and Liu, 2016; Wang et al., 2016; Parracho, 2017; Bastin et al., 2018). These studies reported various levels of agreement between GPS and atmospheric models/reanalyses making it difficult to draw general conclusions on the consistency between products. Indeed, the results depend on the model horizontal and vertical resolution, the method employed (or not employed) for the correction of vertical displacement between the model grid points and stations, and the considered geographical area and period of time. Though the influence of the model horizontal resolution suggests that representativeness differences exist between the model gridded data and station point observations (Lorenc, 1986; Janjić and Cohn, 2006), representativeness errors in IWV data have not so far carefully assessed in these studies. Representativeness differences arise when the station observations capture some small-scale variability that is not resolved by the model/reanalysis. Indeed, model values are representative of spatial averages. Large biases are thus often observed in coastal and mountainous regions (Hagemann et al., 2003; Bock et al., 2005; Parracho et al., 2018). In coastal areas, model grid cells can contain a fraction of IWV over sea not consistent with the GPS observations over land. In mountains, the model IWV can be strongly biased compared to GPS observations made in valleys or uphill. Biases amount typically to -40% IWV per km of height difference (Bock et al., 2005). Since model values are computed above a smoothed orography, which can strongly depart locally from the real topography, a vertical correction is generally applied within a limited altitude range (e.g. ± 500 m). Vertical correction is especially important for variables such as IWV because the water vapour mixing ratio is the largest in the atmospheric boundary layer. Variation of biases/differences between GPS and models is also observed as a function of latitude and season (Roman et al., 2012; Ning et al., 2013; Parracho et al., 2018). Absolute differences have a tendency to be larger in moister and warmer regions/periods while relative differences tend to be larger in colder and drier regions/period, globally. The reasons for these spatial and temporal variations are not clearly understood yet. There are a multitude of possible explanations. For instance, the atmospheric processes can be less well represented in the model/reanalysis (model errors) in some regions and periods of the year, the representativeness differences can be for some unknown reason enhanced, the GPS IWV estimates can have increased errors (e.g. during disturbed/severe meteorological events the mapping function errors would be larger (Boehm et al., 2007)), etc.

The goal of this study is to better understand to which extent model errors, GPS errors, and representativeness errors can be distinguished, what is the limit set by representativeness differences on the best achievable agreement between global reanalyses and station observations, and explain their contribution to the geographical and seasonal dependencies reported in



previous publications. To this purpose we analyse the differences in IWV data from the ECMWF reanalysis, ERA-Interim (Dee et al., 2011), and from a global network of 120 GPS stations (Bock, 2016). We use simple statistics (mean differences and standard deviations, such as found in most past studies) to quantify the consistency between both datasets. We investigate the dependence of these statistics upon latitude, altitude, and time, as well as mean atmospheric moisture content and its spatial

and temporal variability. A representativeness error statistic is introduced which quantifies the spatial variability in the ERA-Interim data at the surrounding grid points and explains partly the observed differences between the reanalysis and the observations. All the statistics are computed over a period of 16 years here because we want to characterize the systematic ERA-Interim minus GPS differences and not their changes over time (e.g. due to inhomogeneity and/or changes in the quality in either of the datasets). The changes over time are small in magnitude (Parracho et al. 2018) and have negligible impact on

the average statistics computed here. After establishing the contribution of representativeness errors, we address the following specific questions: 1) by which means is it possible to mitigate the representativeness errors? 2) does horizontal interpolation of model values degrade or increase their representativeness in comparison to nearby point station observations? 3) can outlying results (e.g. sites with extreme biases and dispersion) be explained as special representativeness errors or are they rather due to model or observation errors? To tackle this question, the seasonal variation of the comparison statistics and of

the atmospheric environment is also analysed. 4) how efficient is the representativeness error statistic in detecting these outlying sites? The results from this study are important to homogenization work where IWV data from reanalyses and GPS observations are used jointly (Vey et al., 2009; Ning et al., 2016; Van Malderen, 2017). Indeed, large representativeness differences put a limit to the use of reanalyses data as a reference for detecting breaks in the GPS time series. Outlying sites should be detected and discarded.

20       The paper is organised as follows. Section 2 describes how the IWV data from the two datasets are prepared. Special effort is made to use a procedure that maximizes the consistency between the datasets. Section 3 presents the results of IWV difference statistics and analyses their dependence upon a variety of parameters. General tendencies are derived that describe the consistency between the reanalysis and GPS globally. Section 4 introduces a range check which detects 15 outlying sites for which the IWV differences are especially large. The geographic, topographic, and seasonal characteristics of these sites is

analysed and site-specific representativeness errors are highlighted. Section 5 discusses the possibility for detecting outlying sites a priori and concludes.

## 2. Data and methods

### 2.1 GPS

In this study we use the tropospheric delay estimates from the first reprocessing of the International GNSS (Global Navigation

Satellite System) Service (IGS), referred to as IGS repro1 (Byun and Bar-Server, 2009; IGSMAIL-6298). It includes results for 456 stations over the period from January 1995 to December 2010. Because we are interested in characterizing the systematic differences between GPS and atmospheric reanalyses, a sub-set of 120 stations which have the longest time series



(16 years) is extracted. The Zenith Tropospheric Delay (ZTD) estimates, which are available with a time sampling of 5 minutes, are first screened for outliers as described in Parracho et al., 2018, and averaged in hourly bins centred on the round hours (00 UTC, 01 UTC…). Next, the hourly ZTDs are converted to IWV using 6-hourly surface pressure, $P_s$, and weighted mean temperature, $T_m$, computed from ERA-Interim pressure level data (see Appendix B in Parracho et al., 2018). No temporal

interpolation is applied here so that only the 1-hourly ZTD estimates matching the times of the reanalysis (00 UTC, 06 UTC…) are converted. Finally, the daily IWV values are computed from five 6-hourly values between 00 UTC of the current day and 00 UTC of the next day, with weights 1/8, 1/4, 1/4, 1/4, 1/8, respectively. Monthly averages are computed directly from the 6-hourly values within the given month to the condition that at least 60 values are available (similar to Parracho et al., 2018). As already mentioned above, inhomogeneities in the GPS IWV time series due to equipment changes are not corrected here. This

does not impact the conclusions since we analyse only overall statistics (means and standard deviations) computed over 16 years but not linear trends. Figure 1 shows the stations used in this study. The GPS coordinates, the altitudes of the reanalysis grid points in the vicinity of the GPS stations, and the number of daily and monthly values for each station are given in the Supplement Table S1.

## 2.2 ERA-Interim reanalysis

ERA-Interim is a modern reanalysis produced by ECMWF using the Integrated Forecasting System (IFS) forecast model and the 4D-Var assimilation system in 12-hourly analysis cycles (Dee et al., 2011). The number of observations has increased from $10^6$ in 1989 per day to $10^7$ per day in 2010. The majority of data, and most of the increase over time, are from satellites. Ground-based GPS data were not assimilated, which make the GPS ZTD and IWV an independent validation dataset. We use ERA-Interim analysis pressure-level data (geopotential, air temperature and specific humidity) extracted from the

Meteorological Archival and Retrieval System (MARS) on a regular latitude-longitude grid with a horizontal resolution of 0.75° x 0.75°. For each and every GPS site, 6-houly ERA-Interim fields are extracted for the four grid points surrounding the GPS station. The IWV contents are computed by integrating the reanalysis specific humidity between the GPS station altitude and the top of atmosphere (1 hPa). For GPS station altitudes located between two pressure levels the ERA-Interim data at the station is interpolated from the adjacent levels. For stations located below the lowest pressure level (1000 hPa), the reanalysis

data is extrapolated. Interpolation and extrapolation are done linearly for specific humidity and temperature, and logarithmically for geopotential, as a function of pressure. To insure the best spatial matching between GPS and ERA-Interim data, the IWV estimates from the four grid points surrounding the GPS station, $IWV_1$ to $IWV_4$, are combined by bi-linear interpolation, resulting in a value denoted by $IWV_{interp}$. Daily and monthly IWV values are computed afterwards in the same manner as for the GPS IWV data (see above).

## 2.3 Comparison method

Daily and monthly time-matched IWV values from GPS and ERA-Interim are compared for each and every station and overall statistics are computed using the full time series (16 years). The overall statistics reveal the systematic or persistent biases and





discrepancies between the two datasets. The goal is to identify the main causes of differences among the representativeness differences, errors in the GPS data, and deficiencies in the reanalysis (e.g. in data-sparse regions). The identification of representativeness differences is made by inspection of a number of statistics and their dependence upon characteristics of the GPS station's environment: moist or dry climate (measured by the mean IWV), strength of temporal variability (measured by

the standard deviation of IWV and of its first derivative), and spatiotemporal variability of IWV in the vicinity of the station. The latter is computed from the ERA-Interim IWV values at the four grid points surrounding the GPS stations. The maximum absolute deviation of the four IWV values, denoted $\delta_{max}IWV$, can reach values as extreme as 18 kg m$^{-2}$ in situations of strong large-scale moisture transport (e.g. in the case of tropical plumes reaching the mid-latitudes). When averaged over one year, the quantity $\mu_R = $ mean $(\delta_{max}IWV)$ is around 2 kg m$^{-2}$ for a typical mid-latitude station and grows up to 6 kg m$^{-2}$ for stations

located in regions of complex topography (e.g. station AREQ in the Andes Cordillera). This quantity is referred to as the "representativeness error statistic" in the following.

All the statistics are defined by equations in Appendix A. The values computed for each station are given in the Supplement Table S2. They may be useful to readers who want to make their own statistical analysis of our results and/or detect outlying sites based on different thresholds than those we used in Section 4.

## 3. Analysis of the general tendency of IWV differences

The mean and standard deviation of IWV differences (ERA-Interim minus GPS) for all 120 stations over the 16-year period are shown in Figs. 2 to 5. Figure 2 shows the results as a function of station latitude. The general tendency is depicted by the fitted polynomials (the outlying stations will be discussed in Section 4). The different plots show a clear dependence of the results on latitude. The mean difference (Figs. 2a, c) is positive at northern and southern extra-tropical latitudes (30-80°N and

30-60°S) while it is negative in the inter-tropical band (30°S – 30°N). This result is consistent with the results of Schröder et al. (2016) who compared ERA-Interim to satellite data. The alternation of positive and negative differences is most likely due to biases in the ERA-Interim reanalysis reflecting the difference in moisture information entering the reanalysis over ocean (mainly microwave satellite data) and land (mainly radiosonde and infrared satellite data) (Dee et al., 2011). Indeed, the tropical GPS stations used here are mostly representative of oceanic areas while the extra-tropical GPS stations are mainly continental.

Similar biases in ERA-Interim were also highlighted by Trenberth et al., 2011, and Parracho et al., 2018, in comparison to other atmospheric reanalyses. The biases remain small, however (below ±1 kg m$^{-2}$ or ± 10%). The absolute standard deviation of IWV differences (Fig. 2b) also shows a latitudinal variation with two peaks, around 30°S and 30°N, and dips around the equator and towards the poles. The equatorial dip is more marked in the relative standard deviation plot (Fig. 2d) because the mean IWV is the largest at these altitudes (~ 40 kg m$^{-2}$, see the blue dashed line in Fig. 2c). The enhanced discrepancy between

ERA-Interim and GPS daily IWV estimates in the sub-tropics coincide quite well with the highest day-to-day variability in both hemispheres (see the superposed blue lines in Figs. 2b, d). This strong day-to-day variability is mainly due to the moisture transport associated with the extra-tropical cyclones in the northern and southern hemisphere storm tracks (Chang et al., 2002;





Pfahl et al., 2014). It is not uncommon to observe IWV variations exceeding 20 kg m$^{-2}$ day$^{-1}$ at GPS sites located in the storm track (Bock et al. 2005; Bock et al. 2016). Increased discrepancy between ERA-Interim and GPS at those sites can be due to the imperfect spatiotemporal location of such large moisture variations in the reanalysis or to a representativeness difference between the GPS observations and the reanalysis. No systematic increase in GPS formal error was found in these situations,

i.e. the discrepancy is not due to GPS errors.

Figure 3 shows the mean and standard deviation of IWV differences as a function of altitude of the GPS stations. The mean differences (Fig. 3a, c) show no dependence on altitude, meaning that the method of computation of GPS IWV (from ERA-Interim $P_s$ and $T_m$ estimates) and ERA-Interim IWV (from pressure levels) are highly consistent throughout a large altitude range. The standard deviation (Fig. 3b) shows no dependence on altitude either but the relative standard deviation (Fig. 3d)

does. The fitted straight line in Fig. 3d shows that this statistic is increasing quite fast as a function of altitude. This tendency can be explained by larger representativeness differences in the reanalysis humidity field as a function of altitude (Waller et al., 2013).

Figure 4 shows the standard deviation of IWV differences, $\sigma_\Delta$, as a function of a few other parameters which give further insight into possible reasons for the discrepancy between GPS and ERA-Interim. Figures 4a and 4b indicate that, apart from

the outliers, there is a moderate tendency for increased discrepancy with increased mean IWV (i.e. warmer and moister climate) and increased IWV variability (including the seasonal variations). The slope of the tendency is actually steeper at the lower IWV bound (mean IWV < 25 kg m$^{-2}$) corresponding to mid and high latitude sites, while it vanishes at the upper bound, corresponding to inter-tropical sites (mean IWV ≥ 25 kg m$^{-2}$). The standard deviation of IWV differences reaches a nearly constant level of $\sigma_\Delta \approx 2$ kg m$^{-2}$ throughout the equator and the inter-tropical band. This finding shows that mid-latitude results

should not be extrapolated towards the equator (a mistake which has been found in several past study and led the erroneous statement that IWV differences increase towards the equator due to the increasing mean IWV). Figure 4c shows that there is a strong tendency for increased discrepancy with increased spatiotemporal variability around the GPS site measured by $\mu_R$ (see Section 2.3). This interrelation is actually the strongest among all the tested relations between $\sigma_\Delta$ and other statistics. It indicates that representativeness differences are a major source of discrepancy between GPS and ERA-Interim IWV estimates.

Finally, Fig. 4d shows that there is only a small tendency for increased discrepancy with increased GPS errors.

Figure 5 shows that time averaging is a means of reducing the representativeness differences, as smaller scale local features captured by the GPS point observations get smoothed out. The mean differences (Figs. 5a, c) are not impacted by the averaging, as expected. The standard deviation of differences (Figs. 5b, d) on the other hand decrease for the monthly averages, both in absolute and relative units, at all sites. The median standard deviation of the daily IWV differences (ERA-Interim minus GPS)

is 1.2 kg m$^{-2}$ while the value for the monthly series is 0.51 kg m$^{-2}$. The reduction of standard deviation due to averaging is 2.35 which is smaller than the value of $\sqrt{30} = 5.48$ that one would expect with independent normally distributed data (when averaging over a mean month of 30 days). This inconsistency can be due to the serial correlation in the IWV differences revealing a dependence of the IWV differences upon the meteorological situation. This point might be further investigated by e.g. separating the IWV differences in different weather regimes. Another means of reducing the discrepancy due to



representativeness differences is to use a reanalysis with higher spatial resolution and improved physics representing the smaller scale atmospheric processes. We compared for instance daily GPS IWV data to the AROME West-Mediterranean operational analysis of Meteo-France (this model has a horizontal resolution of 2.5 km x 2.5 km) and found a median standard deviation of difference of 0.81 kg m$^{-2}$ over a period of 2 months (we used the GPS and AROME data from the HYMEX Special

Observing Period, 5 September – 6 November 2012, described in Bock et al., 2016).

Since representativeness differences impose a strong limitation on the agreement between GPS and reanalysis, one may wonder if the horizontal interpolation from the four surrounding ERA-Interim grid points does not further enhance the differences by mixing information from the different grid points. We investigated this question by computing the statistics for each of the four surrounding grid points. Figure 6 shows the results in comparison to the results obtained with the bi-linearly

interpolated IWV values. The comparison of the mean values (Figs. 6a and 6b) emphasizes large variations in the biases at some stations which will be further discussed in Section 4. The slight shift of the ensemble of results below the 1:1 line is reflecting the fact that a majority of sites exhibit small positive bias (0.47 kg m$^{-2}$ on average) as already noticed in Figs. 2a, c, which is not due to representativeness differences. The comparison of standard deviations (Figs. 6c and 6d) shows unambiguously that at almost all sites, the results for the bi-linearly interpolated IWV values are better than for any one of the

four surrounding grid-points (almost all results sit above the 1:1 line). This conclusion holds for 112 out of 120 stations for the absolute standard deviation (Fig. 6c) and 111 out of 120 stations for the relative standard deviation (Fig. 6d). It indicates that the temporal variability represented by the bi-linearly interpolated ERA-Interim IWV data matches best the temporal variability observed by the GPS (i.e. better than from the nearest grid point in the horizontal or in the vertical dimension). When monthly IWV data are compared (not shown), the conclusions are similar, though the number of sites of improved

results drops to 71 out of 120 (both for absolute and relative standard deviations). The drop confirms again that the representativeness differences can be reduced by the temporal averaging.

## 4. Analysis of outlying sites

In the previous section we have seen that the general agreement between GPS and ERA-Interim is limited by representativeness differences which are enhanced in regions of strong temporal variability (Figs. 2b, d), at higher altitude (where mainly the

relative standard deviation of differences is impacted, Fig. 3d), and at sites where the mean spatial variability at the 4 surrounding ERA-Interim grid points is large (Fig. 4c). The standard deviation of differences, $\sigma_\Delta$, is actually well predicted by our representativeness error statistic, $\mu_R$, with a linear correlation coefficient of $r(\sigma_\Delta, \mu_R) = 0.73$. This strong correlation suggests that the outlying sites, i.e. sites with the largest discrepancy, may have enhanced representativeness errors (Fig. 4c). To investigate this idea, we will analyse in more detail these sites here. First, let us define range limits for each of the four

statistics of differences to separate the acceptable sites (i.e. those satisfying the following conditions) from the outliers:

$$-1 \text{ kg m}^{-2} < \mu_\Delta < 2 \text{ kg m}^{-2}$$

$$-6\% < \mu_\Delta^r < 12\%$$



$$\sigma_\Delta < 2.1 \text{ kg m}^{-2}$$

$$\sigma_\Delta^r < 18\%$$

The values of the limits were determined from visual inspection of Figs. 2, 3, and 5, and shown as the red dotted lines in these figures. The method is subjective, but the chosen values permit to well separate the acceptable from outlying results independently of the latitude and altitude of the sites. The result is a detection of 15 outlying sites, some of which exceed the limits in more than one test: 3 sites have excessive absolute bias (CFAG, KIT3, and SANT); 9 sites have excessive relative bias (CFAG, COSO, DAV1, KIT3, MAW1, MCM4, POL2, SANT, and SYOG); 8 sites have excessive standard deviation of differences (AREQ, BLYT, CFAG, DHLG, IISC, KIT3, LONG, and SANT); and 9 sites have excessive relative standard deviation of differences (AREQ, CFAG, KIT3, MAW1, MCM4, MKEA, POL2, SANT, and SYOG). Three sites have statistics exceeding the limits in all four tests (CFAG, KIT3, and SANT). Two of these sites (CFAG and KIT3) are also characterized by among the largest representativeness error statistics (Fig. 4c).

Figure 7 shows the values of the four comparison statistics for the 15 outlying cases for the bi-linearly interpolated ERA-Interim values and also from the values at the four surrounding grid points (ordered by increasing distance to the GPS station). The results are grouped by region as outlying sites appear to form several clusters located in specific areas of the globe (see Fig. 1). In addition to the four statistics (Figs. 7a to d), we included the altitudes of the GPS stations, $h_{GPS}$, and of the four surrounding grid points (Fig. 7e). The above-chosen range limits are superposed in Figs. 7a to d, and a range limit for the altitudes is indicated as $h_{GPS} \pm 500$ m (Bock et al., 2014).

AREQ , SANT, and CFAG are all three located in the Andes cordillera, with AREQ ($h_{GPS}$ = 2470 m) and SANT ($h_{GPS}$ = 696 m) on the western flank of the mountain range facing the sea, and CFAG ($h_{GPS}$ = 680 m) on its eastern flank. The local topography peaks above 3000 m, 4000 m, and 3000 m within a radius of 100 km from these three sites, respectively. The altitudes of the four surrounding ERA-Interim grid points are very variable (Fig. 7e), and for AREQ (SANT), all (some) of them are exceeding the altitude range limit. At AREQ, absolute and relative standard deviations of the interpolated data exceed slightly the limits, with $\sigma_\Delta$ = 2.4 kg m$^{-2}$ and $\sigma_\Delta^r$ = 21 %, while the bias is almost zero. Moreover, most of the statistics at the four surrounding grid points exceed the range limits. There is thus a significant representativeness difference between the four grid points which is not surprising given the steep orography and strongly varying altitudes of the grid points. Three of the grid points are actually located more than 500 m higher than the GPS station. For these grid points, the validity of the lower pressure level data can be questioned as the atmospheric variables are extrapolated far below the model's surface. The results at SANT have similar issues with biases again correlated with variations in the model topography. At both sites, issues with the GPS measurements were eliminated by verifying their consistency with collocated DORIS measurements (Bock et al., 2014). Compared to AREQ and SANT, CFAG has much worst results and gets actually the worst statistics of all 15 sites: $\mu_\Delta$ = 5.8 kg m$^{-2}$, $\mu_\Delta^r$ = 35 %, $\sigma_\Delta$ = 3.7 kg m$^{-2}$, and $\sigma_\Delta^r$ = 22 %. Contrary to the previous sites, the results for the four grid points are very similar, though the biases vary slightly (from 5.9 to 4.1 kg m$^{-2}$), which suggests that the discrepancy at this site may not be due that much to spatiotemporal variability in the IWV field. Problems with the GPS measurements cannot be excluded at this site and should be checked by comparison with independent observations.





Further insight into the nature of the discrepancies is given by inspection of the seasonal variation of the comparison statistics (Fig. 8) and of the atmospheric environment (Fig. 9). Figure 8 shows that at all three sites, the biases and standard deviations vary over the year, in relation with the variation of the mean IWV ($\mu_W$, Fig. 9a) and the day-to-day variability ($\sigma_W$ and $\sigma_{dW/dt}$, Figs. 9b and 9c, respectively). Both the $\mu_\Delta$ and $\sigma_\Delta$ are peaking when $\mu_W$ is peaking, during the austral summer

months. The relative differences, $\mu_\Delta^r$ and $\sigma_\Delta^r$, and IWV variability, $\sigma_W^r$ and $\sigma_{dW/dt}^r$, are peaking in winter when the mean IWV is low. Inspection of $\mu_R$ (Fig. 9d) confirms the strong impact of spatiotemporal variability at all three sites, but especially at AREQ where it is the largest among all sites (peaking at $\mu_R$= 6.4 kg m$^{-2}$). It is noticeable that at CFAG the yearly mean and the seasonal cycle of IWV in ERA-Interim are larger than observed by GPS (Fig. 9a), which suggests that a representativeness difference is most likely the explanation rather than GPS measurement issues evoked above.

The next two sites, KIT3 ($h_{GPS}$ = 659 m) and POL2 ($h_{GPS}$ = 1755 m), are located in Uzbekistan and Kyrgyzstan, respectively, close to the Alai/Tien Shan mountain range. They both show large difference statistics, with $\mu_\Delta$, $\mu_\Delta^r$, $\sigma_\Delta$, and $\sigma_\Delta^r$ exceeding the limits for KIT3 and $\mu_\Delta^r$ and $\sigma_\Delta^r$ for POL2 (Figs. 7a to 7d). Considering the individual grid points, they almost all also exceed the limits, with large variations both in the bias and standard deviation at KIT3, with somewhat smaller differences at POL2. These variations can again be related to variations in the grid point altitudes, some of which exceed the range limits

(Fig. 7e). The difference statistics at these sites exhibits large seasonal variations, with $\mu_\Delta$, $\mu_\Delta^r$ and $\sigma_\Delta$, peaking in boreal summer (Fig. 8) when $\mu_W$ and $\mu_R$ are peaking (Fig. 9). The representativeness error statistics peaks are particularly marked at these stations, with KIT3 showing the largest monthly values among all sites ($\mu_{R,i}$ = 8.8 kg m$^{-2}$ in August, Fig. 9d). At this site, the GPS measurements were also verified with collocated DORIS measurements (Bock et al., 2014), confirming that representativeness differences between ERA-Interim and GPS IWV data are the main reason for this discrepancy. Interestingly,

it can be noticed that the peak in IWV during summer is significantly larger in ERA-Interim compared to GPS (Fig. 9a), suggesting excessing moisture transport into this region in the reanalysis, possibly connected with the too smooth topography in the model.

The next five sites belong to two geographical regions: IISC, in India, and DHLG, BLYT, LONG, and COSO in California, USA, which are all characterized by small discrepancies with only one statistic exceeding the range limits ($\sigma_\Delta$ for the first four,

and $\mu_\Delta^r$ for COSO). At all five sites, the variation of statistics among the four grid points are small (Figs. 7a to 7d), as are the variations of the altitudes (Fig. 7e). Station IISC shows a small seasonal variation in the bias and standard deviation (Fig. 8) which might be linked to the variation in IWV temporal variability (Figs. 9b, c, e, f) and spatiotemporal variability (Figs. 9d and g) that show peaks in spring and autumn, i.e. during transitions seasons between the summer monsoon (June to October) and the cooler winter season (December to March). It has been shown previously that monsoon transition periods are

accompanied by strong spatial and temporal variability in IWV which is difficult to represent in atmospheric reanalyses (Bock et al., 2008; Bock and Nuret, 2009; Meynadier et al., 2010; Means, 2013).

The four outlying Californian sites can be separated into two groups: DHLG, BLYT, and LONG, located south of the Sierra Nevada mountain range, in a region of moderate topography, and COSO located in the Basin and Range Province, a narrow



valley at the southern exit of the Sierra Nevada. The higher altitude (1485 m) and more complex topographic environment of COSO enhances the representativeness differences. Interestingly, all four sites show a step-like variation of the mean IWV and variability (Figs. 9a, b, c) peaking in July-August-September associated with the North American monsoon (Adams and Comrie, 1997; Means, 2013). This feature is very contrasting with the Indian monsoon observed at IISC where variability was

enhanced during the transition seasons and not during the monsoon. At DHLG and BLYT the biases actually reverse signs in July-August (Figs. 8a, b) and the standard deviation peaks at $\sigma_\Delta > 4$ kg m$^{-2}$ (Fig. 8c). Figure 9b and c show that ERA-Interim underestimates IWV variability at these sites which suggests that GPS observations capture some small-scale moisture variability not represented in the reanalysis.

The next site, MKEA ($h_{GPS}$ = 3730 m) is located on the Mauna Kea volcano on the island of Hawaii. Due to smallness of
the emerged land area (ca 10$^4$ km$^2$), the imprint of the island is almost inexistent in the reanalysis' topography (Fig. 7e). Hence, it is not surprising that the comparison statistics are bad (thought only $\sigma_\Delta^r$ is exceeding the range limits). The relative IWV differences are huge (Figs. 9e and f) when computed with respect to the low GPS IWV content of this high altitude site.

The last group of sites is located in eastern Antarctica (Fig. 1). Unfortunately, four of the five Antarctica sites used in this study suffer from large discrepancies. Three of them have two statistics ($\mu_\Delta^r$ and $\sigma_\Delta^r$) exceeding the range limits (Figs. 7b and
d). MCM4 is the worst case and has the largest relative standard deviation among all 15 sites: $\sigma_\Delta^r = 32\%$. This station is located in McMurdo Detroit, an area with complex landscape, including local low mountain peaks, valleys and glacier corridors, and sea within a radius of 100 km. The other three stations are located close to the coast line backed to the main ice shelf with large surface elevation variations (up to 2000 m within a distance of 100 km). The grid points in ERA-Interim are at different altitudes associated with differences in representativeness leading to IWV biases (Fig. 7b). The marked seasonal variation of
$\mu_\Delta^r$ and $\sigma_\Delta^r$ (Figs. 8b and d) also confirm a dependence of the IWV differences on the atmospheric state and especially on IWV variability which is enhanced during the austral winter months (Figs. 9e and f). The winter variability is actually much underestimated in ERA-Interim as seen in Figs. 9e and f at MCM4, SYOG, and MAW1, and, quite surprisingly, the spatiotemporal variability, $\mu_R^r$, remains nearly constant in ERA-Interim (Fig. 9g). These differences point to an issue in ERA-Interim IWV contents in Antarctica, especially during austral winter, as also suggested by Parracho et al., 2018, who compared
ERA-Interim to the NASA Modern Era Retrospective-Analysis for Research and Applications version 2 (MERRA-2) reanalysis. These authors also pointed to some issues in the GPS measurements at MCM4 and SYOG between 2002 and 2006, as well as a break in the IWV series at all sites in Antarctica due to a discontinuity in the GPS processing. The IWV issues in ERA-Interim may be linked to the large surface air temperature biases of the reanalysis diagnosed by Bracegirdle and Marshall, 2012, from coastal station observations which are related to its too smooth orography. In addition, Xie et al., 2016, showed
that the replicability of daily and annual variance of surface air temperature in this reanalysis decreases from the coast to the interior of the continent. These result also support the findings of Parracho et al., 2018, that the IWV variability and trends in ERA-Interim reanalysis are more realists near the coast where in-situ observations are assimilated than in the interior where the reanalysis mainly relies on satellite observations and short-term model forecasts. Representativeness differences between GPS and ERA-Interim in Antarctica are thus be enhanced by deficiencies in the reanalysis.



## 5. Discussion and conclusions

In this study we first analysed the general tendency of IWV difference between ERA-Interim reanalysis and global GPS observations. We found that the mean difference, interpreted as the bias of the reanalysis with respect to the observations, exhibits a latitudinal variation of $\pm 1$ kg m$^{-2}$, consistent with the fact that different moisture information is entering the reanalysis

over ocean and land. As a result, the northern and southern mid-latitudes exhibit a moist bias, while the tropics are to dry. This bias is not changing with the altitude of the observation site. The standard deviation of daily IWV differences is generally below 2 kg m$^{-2}$ but peaks at the northern and southern storm-tracks latitudes. This result suggests that GPS observations capture some small-scale variability that is not resolved by the reanalysis. Another indication that the discrepancies are process-related is that the relative standard deviation is increasing with altitude (from about 8% at sea level to 16% at 2.5 km). More generally,

it is shown that discrepancies are due to representativeness differences between the gridded reanalysis field and the GPS point observations. A strong correlation (r = 0.73) is found between the standard deviation of IWV differences, $\sigma_\Delta$, and our representativeness error statistic, $\mu_R$, which measures the spatiotemporal variability in the vicinity of the GPS site based on the analysis of the ERA-Interim IWV data at the four surrounding grid points. However, it is shown that in general (for 112 sites out of 120), bi-linearly interpolated IWV values from the four surrounding grid points are in better agreement with the GPS

observations than any of the grid points individually. Even if the horizontal resolution of the reanalysis grid is quite coarse (0.75° x 0.75°), spatial interpolation does not reduce the representativeness. It is also shown that the standard deviation of IWV differences is further reduced when data are time-averaged (e.g. in monthly bins). Indeed, spatial and temporal averaging smooths out the variability due to smaller scale phenomena and make the reanalysis and observations more consistent at representing the larger-scale meteorological systems.

In a second part we analysed in more detail the possible reasons for the very bad comparison results obtained at 15 outlying sites. It is shown that at most of the sites, representativeness errors are the most plausible cause for discrepancies which are enhanced because of local topographic and climatic features. The problematic topographic features include steep orography such as found for sites in the Andes cordillera (AREQ, CFAG, and SANT), on the island of Hawaii (MKEA), close to the Himalayas chain (KIT3 and POL2), as well as coastal sites in Antarctica (MCM4, SYOG, MAW1, and DAV1). The climatic

features include large seasonal changes in the total IWV, such as associated with the Indian monsoon (IISC, KIT3, POL2) or the North American monsoon (DHLG, BLYT, LONG, and COSO), and/or in the IWV synoptic variability (observed at most sites during either the transition seasons, winter, or summer, depending on the geographic location). When these 15 stations are eliminated from the dataset, the comparison statistics become: $\mu_\Delta = 0.36 \pm 0.49$ kg m$^{-2}$, $\mu_\Delta^r = 2.7 \pm 3.5$ %, $\sigma_\Delta = 1.22 \pm 0.38$ kg m$^{-2}$, and $\sigma_\Delta^r = 8.2 \pm 3.0$ % (mean $\pm$ standard deviation over the 105 sites). They are significantly improved compared to the

initial results including the 120 sites: the standard deviations of $\mu_\Delta$ and $\mu_\Delta^r$ are reduced by 30%, the means of $\sigma_\Delta$ and $\sigma_\Delta^r$ by 20% and the standard deviations of $\sigma_\Delta$ and $\sigma_\Delta^r$ are reduced by 40%. Because the comparison of GPS and ERA-Interim is not relevant at these sites, we recommend not to use ERA-Interim in the homogenization process of these GPS time series (Ning et al., 2016; Van Malderen et al., 2017).




These results lead to a more general question whether it is possible to eliminate problematic stations *a priori*, i.e. before the comparison statistics are computed? Inspection of the elevation of the four surrounding grid points with respect to the elevation of the GPS station and with respect to each other provides some indication of possible representativeness errors. Some correlation between IWV biases and altitudes at the individual grid points was found in extreme cases (Fig. 7). A simple

a priori check based on the comparison of grid point altitudes to station altitudes would eliminate some of the problematic cases. We compared the statistics with and without selection of sites where the elevation of the grid points differs by more than 500 m from the GPS station. When the selection is applied to the nearest grid point only, 15 stations are eliminated, including 4 of the outlying sites discussed in Section 4. This test is not very efficient. When applied to all 4 surrounding grid points, 34 stations are eliminated, including 11 of the outlying sites (only CFAG, MCM4, BLYT, and IISC remain then in the

dataset). On average, the statistics of the mean differences ($\mu_\Delta$ and $\mu_\Delta^r$) don't change very much in that case, mainly because the stations with the largest absolute and relative biases (CFAG and MCM4) are not eliminated. However, the statistics of the standard deviation of differences ($\sigma_\Delta$ and $\sigma_\Delta^r$) are reduced by about 20%. However, the benefit is at the expense of a strong reduction of the number of sites (34 stations eliminated). Though altitude differences have been shown to explain discrepancies at certain stations *a posteriori* in Section 4, this altitude check appears too excessive to be applied in a systematic way a priori.

We also tested the use of the absolute and relative representativeness error statistics, $\mu_R$ and $\mu_R^r$, and found that a threshold of 20 % on $\mu_R^r$ eliminates 13 stations, including 8 out of the 15 outlying sites, and reduces the error statistics $\sigma_\Delta$ and $\sigma_\Delta^r$ by 20 to 30 % on average. This outlier check is efficient and is thus recommended. However, none of the checks that we tested was able to detect all the 15 outlying sites. Hence, it is also advised to carefully analyse the comparison statistics in order to understand the possible causes of discrepancies and eliminate outlying stations *a posteriori* on a subjective basis as we have

done in this study. This was possible here because the number of stations was small. In more extended networks, an automatic selection method based on e.g. on a clustering algorithm would be necessary.

Asides from the large representativeness errors found at a small number of sites, one should recognize that ERA-Interim and GPS IWV data are generally in good agreement globally, except perhaps in Antarctica. One of the remaining error sources not addressed in this study is the temporal consistency of both data sources. Therefore, other statistics are more relevant such

trend estimates (Schröder et al., 2016; Parracho et al., 2018). The methodology described in this paper might also help to assess the uncertainties in reanalyses and other observation types.

*Data availability.* GPS IWV data have the following DOI: global GPS IWV data at 120 stations of IGS permanent network, https://doi.org/10.14768/06337394-73a9-407c-9997-0e380dac5591 (Bock, 2016). ERA-Interim data can be downloaded at https://www.ecmwf.int/en/forecasts/datasets/archive-datasets/reanalysis-datasets/era-interim (last access: October 2018; Dee et al., 2011).
The results presented in the paper are also provided in the Supplement.





## Appendix A: Definition of variables and comparison statistics

Throughout this study, the GPS IWV data at a given station is denoted by $IWV_{GPS}$ and the corresponding ERA-Interim IWV data is denoted $IWV_{ERAI}$. When the subscript is not specified, the IWV data may refer interchangeably to GPS and ERA-Interim. When the ERA-Interim IWV data from four surrounding grid points need be distinguished, subscript $i$ is added, with

$i=1..4$, and the bi-linearly interpolated value is then denoted by $IWV_{interp}$.

GPS and ERA-Interim IWV data are analysed using the following statistics, where the mean and standard deviation are computed over the number of days (months) of the time-matched daily (monthly) data:

- The mean and standard deviation of IWV:

$$\mu_W = mean(IWV) \tag{A1}$$
$$\sigma_W = std.dev.(IWV) \tag{A2}$$

- The relative standard deviation of IWV:

$$\sigma_W^r = \frac{std.dev.(IWV)}{mean(IWV)} \tag{A3}$$

- The standard deviation and relative standard deviation of IWV time derivate:

$$\sigma_{dW/dt} = std.dev(dIWV/dt) \tag{A4}$$
$$\sigma_{dW/dt}^r = \frac{std.dev(dIWV/dt)}{mean(IWV)} \tag{A5}$$

The ERA-Interim representativeness error statistic is based on the maximum absolute difference in IWV from the four surrounding grid points, $\delta_{max}IWV = \max_i(IWV_{ERAI,i}) - \min_i(IWV_{ERAI,i})$:

- The absolute and relative mean "representativeness error statistic":

$$\mu_R = mean(\delta_{max}IWV) \tag{A6}$$
$$\mu_R^r = \frac{mean(\delta_{max}IWV)}{mean(IWV_{ERAI,interp})} \tag{A7}$$

The ERA-Interim minus GPS differences are analysed using the following statistics:

- The mean and standard deviation of IWV differences:

$$\mu_\Delta = mean(IWV_{ERAI} - IWV_{GPS}) \tag{A8}$$
$$\sigma_\Delta = std.dev.(IWV_{ERAI} - IWV_{GPS}) \tag{A9}$$
- The relative mean and standard deviation of IWV differences:

$$\mu_\Delta^r = \frac{mean(IWV_{ERAI} - IWV_{GPS})}{mean(IWV_{GPS})} \tag{A10}$$

$$\sigma_\Delta^r = \frac{std.dev(IWV_{ERAI} - IWV_{GPS})}{mean(IWV_{GPS})} \tag{A11}$$

In equations (A6) to (A9), the ERA-Interim IWV values can be $IWV_{interp}$ (as in Figs. 2 to 5) or any one of the $IWV_{ERAI,i}$ when individual grid points are discussed (as in Figs. 6 and 7). In Section 4 of the manuscript, statistics from Fig. 7 are referred

to $\mu_{R,i}, \mu_{R,i}^r \ldots$ when the representativeness error estimates from individual grid points are discussed.





The units of the values computed using Eqs. (A1, A2, A6, A8, A9) is kg m$^{-2}$.

The units of the values computed using Eqs. (A3, A10, A11) is % when multiplied by 100.

The units of the values computed using Eq. (A4) is kg m$^{-2}$ day$^{-1}$ and for Eq. (A5) it is % day$^{-1}$ when multiplied by 100.

*Supplement.* This article includes 2 supplement tables.

*Author contributions.* OB prepared the GPS and ERA-Interim data, performed the comparisons, and wrote the paper. ACP contributed to the data analysis and discussion of results.

*Competing interests.* The authors declare that they have no conflict of interest.

*Special issue statement.* This article is part of the special issue "Advanced Global Navigation Satellite Systems tropospheric products for monitoring severe weather events and climate (GNSS4SWEC) (AMT/ACP/ANGEO inter-journal SI)". It is not associated with a conference.

*Acknowledgments.* This work was developed in the framework of the VEGA project and supported by the CNRS program LEFE/INSU. This work is a contribution to the European COST Action ES1206 GNSS4SWEC (GNSS for Severe Weather and Climate monitoring; http://www.cost.eu/COST_Actions/essem/ES1206) aiming at the development of the global GPS network for atmospheric research and climate change monitoring. This study benefited from the IPSL mesocenter ESPRI facility which is supported by CNRS, UPMC, Labex L-IPSL, CNES and Ecole Polytechnique which is funded by the ANR (Grant #ANR-10-LABX-0018) and by the European FP7 IS-ENES2
project (Grant #312979).

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





**Figures**

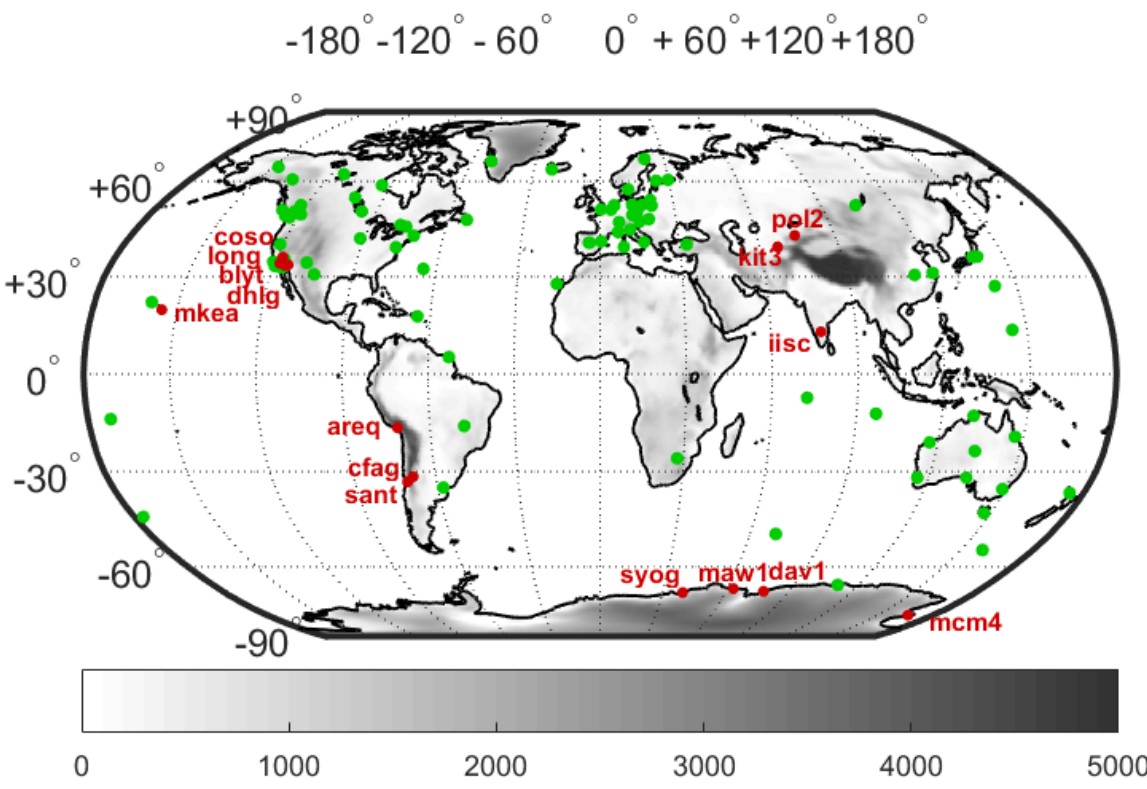

**Figure 1.** Map showing the 120 GPS stations used in this study. A dynamic map including geographical and technical
information for all the GPS sites can be found on http://www.igs.org/network. Outlying sites (named in red) are detected using
a range check based on IWV difference statistics (ERA-Interim minus GPS). The grey shading shows the surface elevation
represented in ERA-Interim, from 0 to 5000 m.



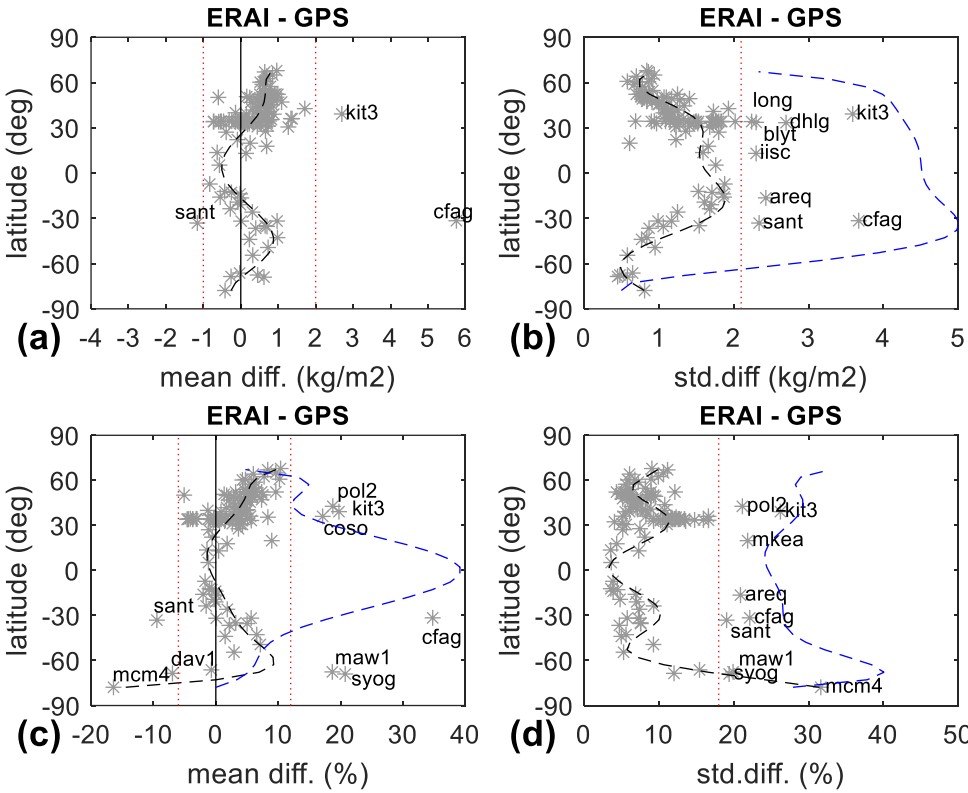

**Figure 2. (a, b)** Mean and **(c, d)** standard deviation of daily IWV difference (ERAI minus GNSS) for 120 global stations as a function of station latitude. **(a, b)** in kg m$^{-2}$; **(c, d)** in % of GNSS IWV. The black dashed lines show polynomial fits of order 5 and 9 for the mean difference and the standard deviation, respectively. The blue dashed lines show polynomial fits of order 7 for **(b)** the standard deviation of dIWV/dt (kg m$^{-2}$ day$^{-1}$); **(c)** the mean IWV (kg m$^{-2}$); **(d)** the relative standard deviation of dIWV/dt (% day$^{-1}$) computed from GPS IWV data. The red dotted lines show the range-check limits used to detect outlying sites (named stations).




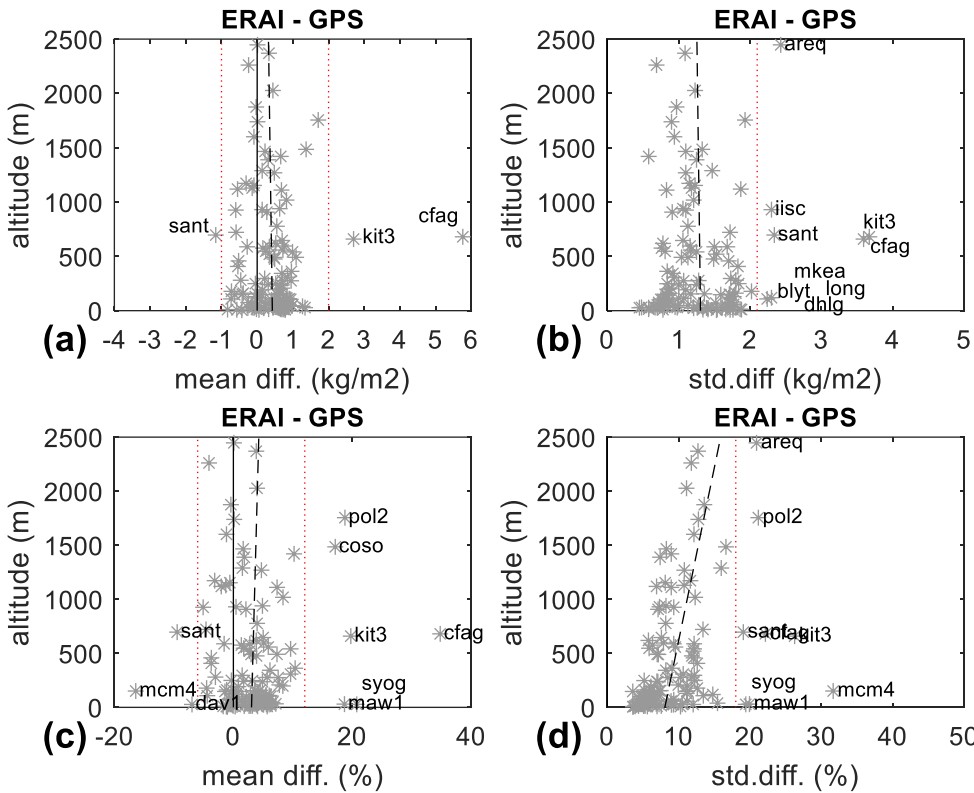

**Figure 3.** Similar to Fig. 2 but plotted as a function of GPS station altitude.





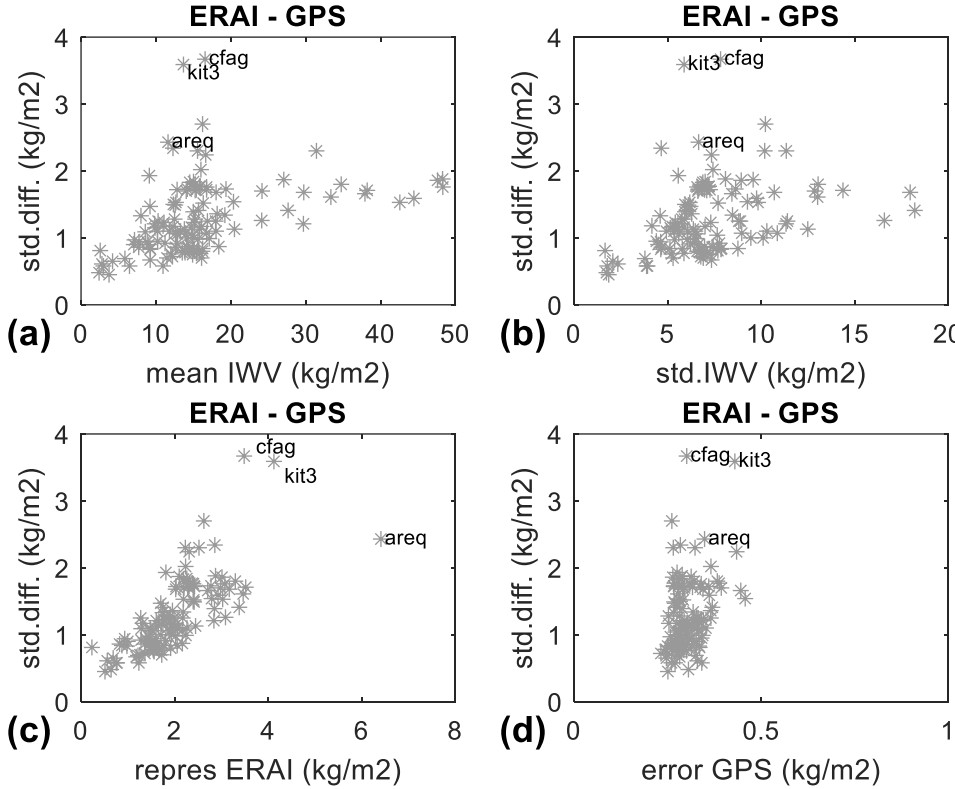

**Figure 4.** Standard deviation of daily IWV difference (ERAI minus GNSS) for 120 global stations, as a function of **(a)** mean GPS IWV; **(b)** standard deviation of GPS IWV; **(c)** mean spatial variability of ERAI IWV from the 4 grid-points surrounding the GPS sites used as representativeness statistic (see text); **(d)** formal error of GPS IWV estimates. Only three outlying stations

5    are named on these plots for clarity.





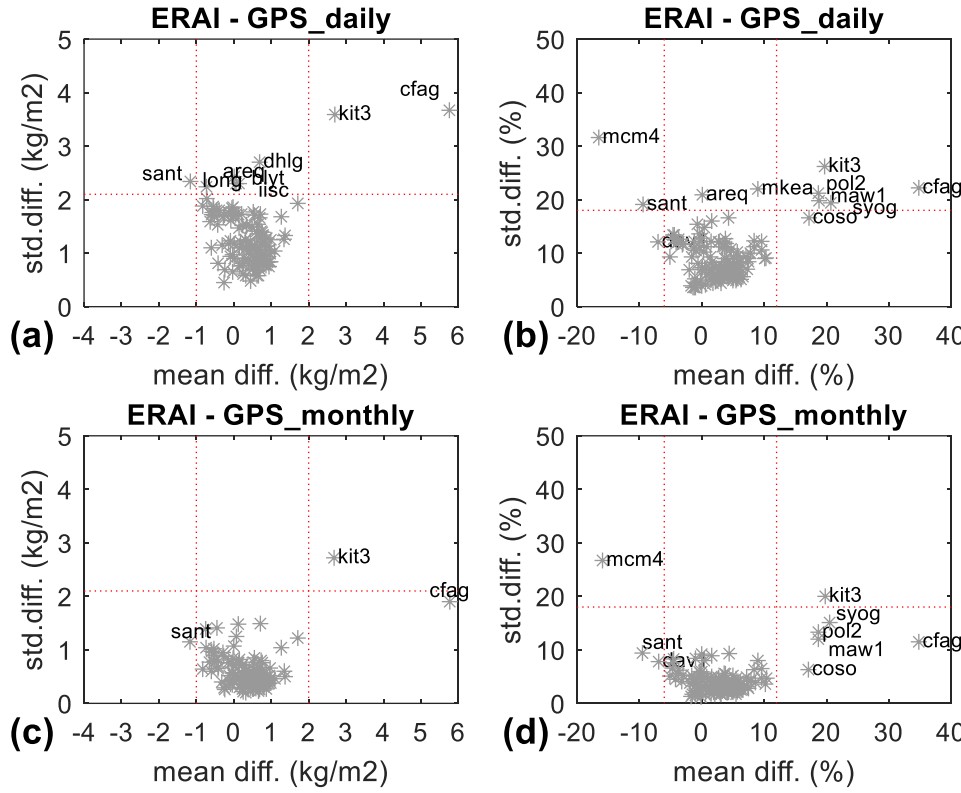

**Figure 5.** Mean vs. standard deviation of IWV difference (ERAI minus GNSS) for **(a, b)** daily values and **(c, d)** monthly values. The median values of mean and standard deviation over all 120 stations are: 0.47 kg m$^{-2}$ and 1.2 kg m$^{-2}$ (3.1 and 8.3 %), for the daily results, and 0.47 kg m$^{-2}$ and 0.51 kg m$^{-2}$ (3.1 and 3.8 %) for the monthly results, respectively. The red dotted lines show the range-check limits used to detect outlying sites (named stations) in the case of the daily data.





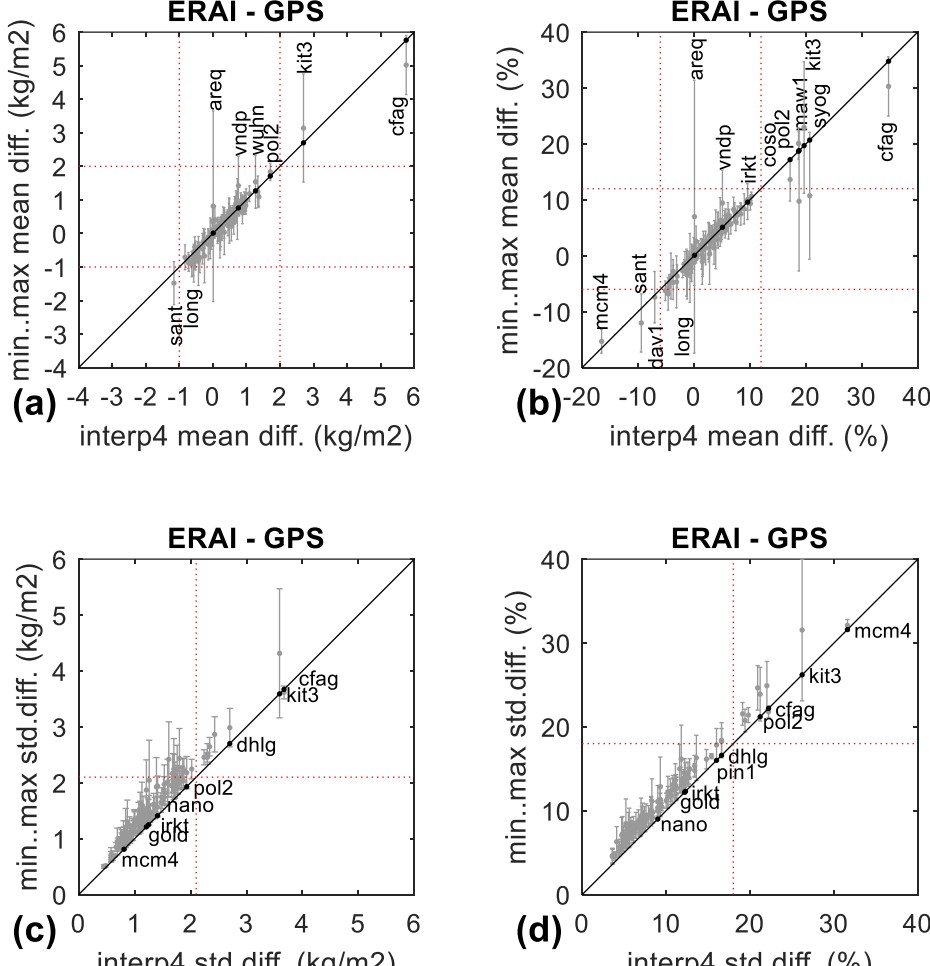

**Figure 6.** Scatter plots of **(a, b)** mean and **(c, d)** standard deviation of daily IWV difference (ERAI minus GPS) when ERAI

5    IWV is bi-linearly interpolated from 4 surrounding grid-points (x-axis) versus the spread of the mean **(a, b)** or standard

deviations **(c, d)** for the four surrounding grid-points (y-axis). The spread is plotted as vertical error-bars from the minimum

to maximum values. In **(c, d),** vertical bars extending below the 1:1 line indicate sites where at least one of the surrounding

grid-points is in better agreement with GPS than the bi-linearly interpolated values; the corresponding stations are named and

indicated by a black dot. The red dotted lines show the range check limits as in previous figures. In **(a, b)**, some of the sites

10   with statistics outside the limits indicated by the red dotted lines are named as well.





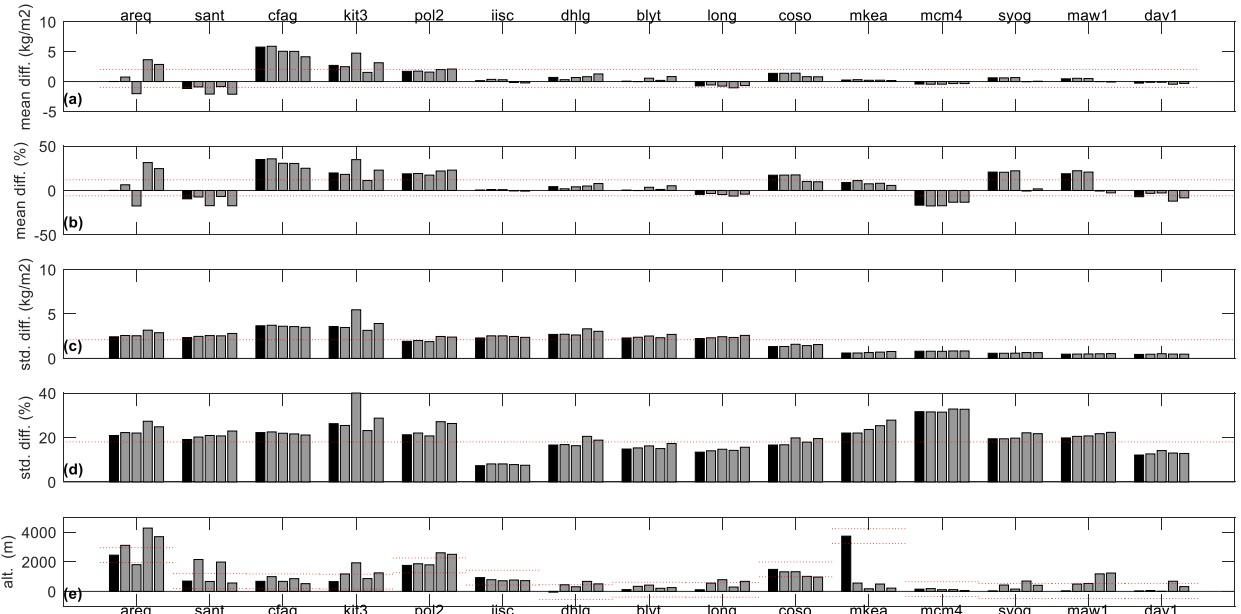

**Figure 7. (a, b)** Mean and **(c, d)** standard deviation of daily IWV difference (ERAI minus GNSS) for 15 outlying sites grouped by region: Andes (cfag, sant, areq), Central Asia (kit3, pol2), India (iisc), western USA (dhlg, blyt, long, coso), Hawaii (mkea), and Antarctica (mcm4, syog, maw1, dav1). In plots **(a)** to **(d)**, the black bars show results for the bi-linearly interpolated ERA-Interim data, and the grey bars the results for the four surrounding grid points, ordered by increasing horizontal distance from the GPS station. Plot **(e)** shows the altitudes of the GPS stations (black bar) and the altitudes of the four surrounding grid points (grey bars). The red dotted lines show the acceptable range limits, similar to Fig. 2 for plots **(a)** to **(d)**, and ± 500 m around the GPS station's altitude in plot **(e)**.





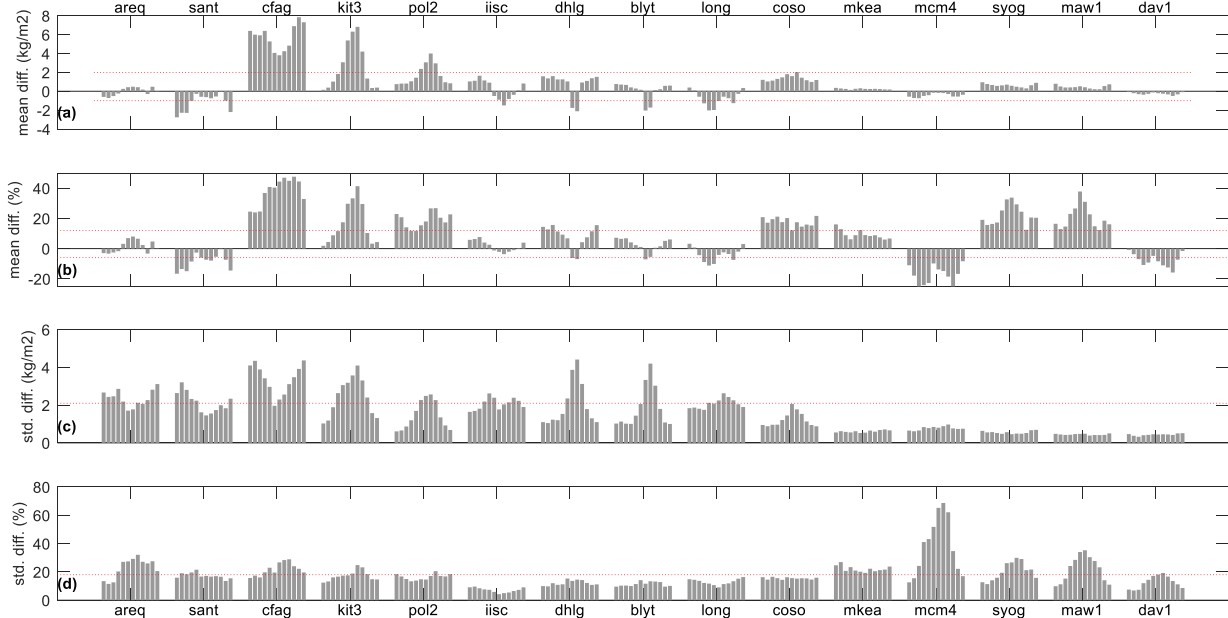

**Figure 8.** Seasonal variation of **(a, b)** mean and **(c, d)** standard deviation of daily IWV difference (ERAI minus GNSS) for 15 outlying sites. The grey bars show the statistics computed for each month (January to December, from left to right) over the 16-year period. The red dotted lines show the range check limits.



**Figure 9.** Seasonal variation of daily IWV data: **(a)** mean IWV; **(b, e)** absolute and relative standard deviation of IWV; **(c, f)** absolute and relative standard deviation of IWV derivative; **(d, g)** absolute and relative mean spatial variability of ERAI IWV from the 4 grid-points surrounding the GPS sites. The grey (blue) bars show GPS (ERA-Interim) statistics computed for each month (January to December, from left to right) over the 16-year period.