# Peer review of "Consistency and representativeness of integrated water vapour from ground-based GPS observations and ERA-Interim reanalysis"

_Atmospheric Chemistry and Physics, 2019_

## Referee Comment (RC1) · Anonymous Referee #1 · 27 Mar 2019

2019-03-26

Review of

*Consistency and representativeness of integrated water vapour from ground-based GPS observations and ERA-Interim reanalysis*

Olivier Bock and Ana C. Parracho

Atmos. Chem. Phys. Discuss., https://doi.org/10.5194/acp-2019-28

**General Comments**

The work presented focus on comparing atmospheric integrated water vapour (IWV) estimated from ground-based GPS observations with the corresponding IWV values in four nearby grid points of the ERA Interim reanalysis. The structure of the manuscript is straightforward and reasonably easy to follow, although I needed quite some time until I was familiar with the nomenclature and the symbols.

The stated motivation for the work was to identify GPS stations where ERA Interim is not recommended to be used when searching for inhomogeneities in the GPS time series of IWV.

A question that is not answered after reading the manuscript is an approximate quantitative relation between representativeness statistics and the size of the break in the GPS IWV time series. I think like this: if representativeness errors (at a specific GPS site) are stable with time, it should still be possible to detect a break in the GPS time series if it is above a certain size? It would be interesting to have the authors ideas about how large, or small, breaks that could be detected, given some example values of the representativeness statistics.

Figures 2–6 are presented and discussed in Section 3. Some of them have red dotted lines defining limits in order to identify outlying results/stations. However, it is only in Section 4 that these limits are explained. I think it would help the reader if they were introduced already in Section 3. Related to this it is clearly stated that the method is subjective. Nevertheless, if the method is to be applied by others, it would be informative to also document the reasoning behind the choices. For

example, why did you choose non-symmetric limits for the mean differences in Figures 2 and 3?

**Specific comments**

P1,L19: It is not surprising that the comparison results are significantly improved when the worst 15 sites (of 120 sites) are removed. It would be informative to quantify the improvement.

P2,L25-26: Is that not obvious? I mean it is stronger than "a tendency".

P2,L32-34: Are representativeness errors never to be referred to model errors? I interpret the definition of a representativeness error as that the only cause is the limited model resolution? If this is correct it can be stated explicitly, because I can also argue that the limited resolution of a model can be the cause of "model errors".

P3,L15: Explain/give examples, what is meant by "atmospheric environment" already here? Although it is clarified later when presenting Figure 9, my reading stopped here wondering what atmospheric environment could be out of many different things?

P4,L9-11: I think you should mention that the GPS time series used have passed some kind of quality check, because a very large break should have an impact on the overall standard deviation of the differences GPS – ERA Interim.

P6,L5: It cannot be taken for granted that the discrepancy is not due to GPS errors just because the formal errors do not increase. For example, a nearby installation of say a metallic structure may introduce significant multipath errors without affecting the formal errors.

P6,L30-33: An additional explanation could be that you only required 15 days of data for a specific month in order to be included. That would also affect the reduction of the standard deviation, unless it is very rare that so much data are missing from a month?

P7,L2-5: Can you compare this standard deviation of 0.81 kg/m$^2$ to what is obtained for stations located in the same area of the present study, in order to quantify the improvement obtained for the higher resolution model?

P7,L13-15: Perhaps the GPS sites that do not show an improvement using bi-linear interpolation are located close to one of the four grid points that is more representative compared to the others?

P11,L11: delete "strong" and just give the value? It should be up to the reader to decide what is a strong and a weak correlation.

P12,L23: delete "good", or state your definition for "good". Which parameter values do you typically see for Antarctica that is not seen globally?

P12,L25-26: This last sentence is not clear. It is the word "also" that raise questions. Because isn't that what you have done in the study? And what is meant by "other observation types"

Fig. 3: The figure caption refers to Figure 2. Are really the black dashed lines in Figure 3 of order 5 to 9?

**Technical Corrections**

P1,L7: IWV is not defined

P1,L17: don't $\Rightarrow$ do not (style ? + a couple of additional ones in the manuscript)

P1,L18: topography and coast-lines, strong $\Rightarrow$ topography, coast-lines, and a strong

P3,L7: delete "here"

P3,L24: sites is $\Rightarrow$ sites are

P6,L20: past study and led the $\Rightarrow$ past studies and led to the

P6,L25: GPS errors $\Rightarrow$ GPS formal errors?

P8,L25: strongly varying $\Rightarrow$ (very) different (altitudes of specific sites do not vary?)

P8,L30: worst $\Rightarrow$ worse

P9,L21: excessing $\Rightarrow$ excessive ?

P10,L11: thought $\Rightarrow$ though (or although?)

P10,L32: realists $\Rightarrow$ realistic ?

P12,L24-25: such trend $\Rightarrow$ such as trend

All figures: the red dotted lines would benefit from having larger dots

Fig. 2: (a,b) Mean and (c,d) $\Rightarrow$ (a,c) Mean and (b,d)

Fig.2: The units specified in the caption for graphs (b) and (c) do not agree with the labels.

Fig. 2: It is difficult to see the difference between the blue and the black dashed lines. Black and green may be better? Or make one of them dash-dotted?

Fig. 5: It is hard to read "dav1"(?) in graphs (b) and (d), plot labels after symbols?

---

## Referee Comment (RC2) · Anonymous Referee #2 · 28 Apr 2019

The manuscript examines the consistency and representativeness differences of daily IWV data from ERA-Interim reanalysis and GPS observations at 120 global sites. The differences are analyzed in details by correlating with various factors and developing a representativeness error statistic using the reanalysis values over the four grid points surrounding the GPS station. The study itself has some values although the scientific originality and applications are not very appealing. Minor comments: 1. I think that one of main motivations for studying the spatial representativeness error is to provide such information to the data assimilation. This should be mentioned in the introduction

along with the current knowledge of the spatial representativeness error of the GPS-derived PWV and whether and how this study tackles this issue. For example, Liou et al. (2001, Comparison of precipitable water observations in the near tropics by GPS, microwave radiometer, and radiosondes. J. Appl. Meteor., 40, 5–15) discussed the sampling differences among different measurement techniques. 2. Page 3, L17-19: "Indeed, large representativeness differences put a limit to the use of reanalyses data as a reference for detecting breaks in the GPS time series. Outlying sites should be detected and discarded." This is a very good point. Comparing the point measurement with the reanalysis has been often used for homogenization. It would be useful to have some results or at least more discussions on this application. 3. Page 4, L21: averaging the values from the surrounding four grid boxes has been used in Mears et al. (2014, Intercomparison of total precipitable water measurements made by satellite-borne microwave radiometers and ground-based GPS instruments, J. Geophys. Res. Atmos., 120, doi:10.1002/2014JD022694) for satellite data.

---

## Author Comment (AC1) · 3 Jun 2019

**Answers to Referee n°1**

We thank the referee for the taking the time to review this paper and providing constructive comments and suggestions. The referee's comments are repeated below in black italics and our answers are given in blue.

*General Comments*

*The work presented focus on comparing atmospheric integrated water vapour (IWV) estimated from ground-based GPS observations with the corresponding IWV values in four nearby grid points of the ERA Interim reanalysis. The structure of the manuscript is straightforward and reasonably easy to follow, although I needed quite some time until I was familiar with the nomenclature and the symbols.*

Thank you for the comment. You are right that we used a number of symbols to quantify our results in the text (rather than paraphrasing) but they were all defined in the Appendix. This may actually require some back and forth reading but the advantage of an Appendix is that all the definitions are grouped in one place.

*The stated motivation for the work was to identify GPS stations where ERA Interim is not recommended to be used when searching for inhomogeneities in the GPS time series of IWV.*

Actually the goal of the study was stated as "to better understand to which extent model errors, GPS errors, and representativeness errors can be distinguished, what is the limit set by representativeness differences on the best achievable agreement between global reanalyses and station observations, and explain their contribution to the geographical and seasonal dependencies reported in previous publications." (page 2 line 32 to page 3 line 1).

Later we mention the application of these results to homogenization: "The results from this study are important to homogenization work where IWV data from reanalyses and GPS observations are used jointly…" (page 3 line 16-17).

*A question that is not answered after reading the manuscript is an approximate quantitative relation between representativeness statistics and the size of the break in the GPS IWV time series. I think like this: if representativeness errors (at a specific GPS site) are stable with time, it should still be possible to detect a break in the GPS time series if it is above a certain size? It would be interesting to have the authors ideas about how large, or small, breaks that could be detected, given some example values of the representativeness statistics.*

This is an interesting question but it would require to establish a rule between our representativeness statistic and the results from a homogenization method. So this is clearly beyond the scope of this paper. Our recommendation here is simply to discard the stations that were detected as outliers based on the proposed thresholds for this dataset (page 11 line 31-33).

Actually, it is difficult to give a more general answer. The size of breaks that can be detected depends strongly on the statistical test or homogenization method used. One of the problems is namely that representativeness errors show in general a strong seasonal variability (see Fig. 8 and 9). In this respect, we think it is primordial that the homogenization method takes the non-stationarity of the variance into account.

*Figures 2–6 are presented and discussed in Section 3. Some of them have red dotted lines defining limits in order to identify outlying results/stations. However, it is only in Section 4 that these limits are explained. I think it would help the reader if they were introduced already in Section 3. Related to this*

*it is clearly stated that the method is subjective. Nevertheless, if the method is to be applied by others, it would be informative to also document the reasoning behind the choices. For example, why did you choose non-symmetric limits for the mean differences in Figures 2 and 3?*

Regarding the outlying stations, we actually refer the reader to Section 4 when we present the first figure in Section 3 (page 5 line 18). To make it clear that the red dotted lines are related to the outlying sites we clarified the sentence in brackets:

"(the outlying stations, defined beyond the red dotted lines, will be discussed in Section 4)."

The reason why we added the red dotted lines and named the outlying results on the figures presented in Section 3 is that this avoids us to duplicate the figures later in Section 4.

The choice for the limits is subjective because we think that the results are very unpredictable when analysing a global network (due to the variety of climates, equipment, and reanalysis performance). So it is necessary to inspect visually the results and determine the limits beyond which the results do not look "normal". We believe this approach is quite robust thanks to the combination of several representations of the results such as shown in Figure 2-6 (i.e. function of latitude, altitude, mean vs. std. scatter plots, etc.). This methodology can be safely applied to other datasets.

The reason why we chose non-symmetric limits wrt to zero for the mean differences is because the distribution is not centred on zero. This is quite clear in Fig. 2, 3, and 5. Choosing symmetric limits here would remove more stations on one side, which is not wanted.

*Specific comments*

*P1,L19: It is not surprising that the comparison results are significantly improved when the worst 15 sites (of 120 sites) are removed. It would be informative to quantify the improvement.*

It is quantified in the next sentence (20 to 30%).

*P2,L25-26: Is that not obvious? I mean it is stronger than "a tendency".*

Well, many studies reported the same ("Absolute differences have a tendency to be larger in moister and warmer regions/periods while relative differences tend to be larger in colder and drier regions/period, globally.") and some hypotheses have been made about the reasons but no clear explanation has been found. So, though the conclusions are not new, it doesn't make them obvious as long as they are not fully explained.

Actually, the dependence of absolute and relative errors on IWV and other factors depends on the observation/processing method and the nature of the noise/error sources. E.g. lidar water vapour measurements errors follow different statistics.

*P2,L32-34: Are representativeness errors never to be referred to model errors? I interpret the definition of a representativeness error as that the only cause is the limited model resolution? If this is correct it can be stated explicitly, because I can also argue that the limited resolution of a model can be the cause of "model errors".*

We distinguish model errors (i.e. error due to the model physics) and representativeness errors (more directly linked to the model resolution and the fact that it cannot represent small scales that are sensed by the observing system). We think the difference is clearly stated page 2 line 15-17. But you are right, the limited model resolution can also be a cause for model errors (e.g. when convection is parameterized vs. explicit). This is one of the reasons why better results are found with the AROME model (page 6 line 25 – page 7 line 5).

It is actually difficult to discuss further the model errors in this study because we have no diagnostic to evaluate them contrary to the representativeness error (according to our definition) for which we proposed a statistic. This statistic can be computed easily from the IWV at the four surrounding grid points and can be used to detect outlying sites, e.g. for the purpose of GPS IWV homogenization.

*P3,L15: Explain/give examples, what is meant by "atmospheric environment" already here? Although it is clarified later when presenting Figure 9, my reading stopped here wondering what atmospheric environment could be out of many different things?*

We added "(mean IWV and variability)"

*P4,L9-11: I think you should mention that the GPS time series used have passed some kind of quality check, because a very large break should have an impact on the overall standard deviation of the differences GPS – ERA Interim*

We added "and the data have been screening beforehand".

*P6,L5: It cannot be taken for granted that the discrepancy is not due to GPS errors just because the formal errors do not increase. For example, a nearby installation of say a metallic structure may introduce significant multipath errors without affecting the formal errors.*

In general, when the noise in the measurements is increasing the formal errors are increasing too because they are rescaled based on the "a posteriori variance factor". So we would expect that a sudden increase in multipath error would be detected by the screening procedure, though such a case has not yet been clearly identified. On the other hand, since we compute statistics over 16 years it is likely that an undetected temporary increase in multipath errors would not impact strongly our statistics.

*P6,L30-33: An additional explanation could be that you only required 15 days of data for a specific month in order to be included. That would also affect the reduction of the standard deviation, unless it is very rare that so much data are missing from a month?*

In this study "Monthly averages are computed directly from the 6-hourly values within the given month to the condition that at least 60 values are available (similar to Parracho et al., 2018)." (page 4 line 7-8). We don't think this can impact the statistics of differences because the GPS and ERAI data are time-matched before the monthly values are computed for each dataset. So there is no sampling difference.

*P7,L2-5: Can you compare this standard deviation of 0.81 kg/m2 to what is obtained for stations located in the same area of the present study, in order to quantify the improvement obtained for the higher resolution model?*

There are 12 IGS stations in the AROME-WMED domain. The median standard deviation of IWV differences GPS-ERAI over these 12 stations amounts to 0.98 kg m$^{-2}$. This number is indeed larger than the 0.81 kg m$^{-2}$ found with AROME-WMED (over 661 stations), but the difference is not only due to resolution but also to different and more modern model physics in AROME-WMED (in addition to different spatial sampling 661 vs. 12 stations).

*P7,L13-15: Perhaps the GPS sites that do not show an improvement using bi-linear interpolation are located close to one of the four grid points that is more representative compared to the others?*

This hypothesis can be tested from Figure 7 where the grey bars shows the results for each of the four grid points ordered by increasing horizontal distance from the GPS station. At 8 out of 15 sites, the closest grid point gives the smallest std. dev. of difference, so about 50% of the cases.

*P11,L11: delete "strong" and just give the value? It should be up to the reader to decide what is a strong and a weak correlation*

Given the different nature of the two variables compared we think that the 0.73 value can be considered as strong (this is a different situation from the comparison of a similar variable from two difference data sources where we would require at least a value of 0.90 to be strong). And in general, we think authors should give their interpretation of results and not leave the task to the reader.

*P12,L23: delete "good", or state your definition for "good". Which parameter values do you typically see for Antarctica that is not seen globally?*

In Antarctica, the comparison statistics exceed the global thresholds at 4 out of 5 sites (see Fig. 2 for the values of the thresholds). We completed the sentence to clarify this point:

"where the comparison failed at 4 sites out of 5."

*P12,L25-26: This last sentence is not clear. It is the word "also" that raise questions. Because isn't that what you have done in the study? And what is meant by "other observation types"*

You are right the sentence was not clear. We reformulated it:

"The methodology described in this paper can also be applied to assess the consistency and representativeness of other data sources (e.g. climate models, satellite IWV data) and other observation types (e.g. surface humidity, temperature, etc.)."

*Fig. 3: The figure caption refers to Figure 2. Are really the black dashed lines in Figure 3 of order 5 to 9?*

You are right, they are linear fits. We added the information in the captions.

*Technical Corrections*

Thank you for the careful reading. All the suggested corrections have been implemented in the text.

Figures: the red dotted lines are kept thin because we don't want to highlight them when we discuss the overall results in Section 3. Later in Section 4, we focus on the named station results, so again the thresholds do not need to be emphasized too much.

An example with linewidth=1 is given below and we think it emphasizes too much the red dotted lines.

We made the blue lines dash-dotted as suggested in Fig. 2 and moved slightly the station labels in Fig. 5 (for dav1 and a few other stations).

[Figure]

---

## Author Comment (AC2) · 3 Jun 2019

**Answers to Referee n°2**

We thank the referee for the taking the time to review this paper and providing short comments. The referee's comments are repeated below in black italics and our answers are given in blue.

*The manuscript examines the consistency and representativeness differences of daily IWV data from ERA-Interim reanalysis and GPS observations at 120 global sites. The differences are analyzed in details by correlating with various factors and developing a representativeness error statistic using the reanalysis values over the four grid points surrounding the GPS station. The study itself has some values although the scientific originality and applications are not very appealing.*

We think the study is important to the GNSS community according to our experience and discussions within the COST Action GNSS4SWEC, and thus fits well into the scope of this Special Issue.

*Minor comments:*

*1. I think that one of main motivations for studying the spatial representativeness error is to provide such information to the data assimilation. This should be mentioned in the introduction along with the current knowledge of the spatial representativeness error of the GPS derived PWV and whether and how this study tackles this issue. For example, Liou et al. (2001, Comparison of precipitable water observations in the near tropics by GPS, microwave radiometer, and radiosondes. J. Appl. Meteor., 40, 5–15) discussed the sampling differences among different measurement techniques.*

We agree with the referee and added the following sentence near the end of the Introduction:

"This study may also contribute to a better treatment of ground-based GNSS observation error in data assimilation, e.g. by establishing a parametric model of observation error depending on the spatio-temporal variability of IWV around the GNSS site computed from the model fields."

The current knowledge of the spatial representativeness error of the GPS derived PWV is actually poor as mentioned P2L15.

*2. Page 3, L17-19: "Indeed, large representativeness differences put a limit to the use of reanalyses data as a reference for detecting breaks in the GPS time series. Outlying sites should be detected and discarded." This is a very good point. Comparing the point measurement with the reanalysis has been often used for homogenization. It would be useful to have some results or at least more discussions on this application.*

This is one of the motivations of this work as also highlighted by Referee n°1. So far, we propose a method to detect the cases where ERA-Interim reanalysis cannot be used to compute reliable IWV differences with the GPS time series. We cannot go further into this discussion for now as explained in the answer to Referee n°1.

*3. Page 4, L21: averaging the values from the surrounding four grid boxes has been used in Mears et al. (2014, Intercomparison of total precipitable water measurements made by satellite borne microwave radiometers and ground-based GPS instruments, J. Geophys. Res. Atmos., 120, doi:10.1002/2014JD022694) for satellite data.*

We don't understand the goal of the referee's comment. Actually, Mears et al., use a slightly different method. They make a bilinear fit of the IWV field from the gridded satellite data (the fit is made in a 7x7 grid cell region surrounding the GPS station). In our case, we use a bilinear interpolation based on the four surrounding grid points but nowhere we write that this method is new. Indeed, it has been used in many past studies. What is maybe original in our study is that we

compare the GPS data to both the interpolated values and to the values of the four grid points. From there we conclude that in general the representativeness of the interpolated values is higher (P7L6-L21).

---

## Author Response (AR2)

**Answer to Co-Editor Decision**

We thank the co-editor for managing our manuscript and providing additional comments that we think helped sharpen the message, especially regarding the concept of representativeness errors. The comments are repeated below in black font and our answers are given in blue.

5    thanks for posting your authors comments and providing an updated version of the manuscript. You mostly addressed in your comments the issues raised by the 2 reviewers, but I would like to have those answers more included in the manuscript itself. After all, most readers will only read the manuscript.

We indeed answered all the referee comments in the previous response letter, but it is true that we did not implement all of them in the manuscript. The reason is that we felt that not all of them were calling for modifications of the manuscript.

10    This certainly applies to the comments raised by both reviewers about the applicability of the method for homogeneity checks of the datasets. Based on the log-files of the different IGS stations, some obvious breaks can be detected in the time series and have already been reported for some stations. Therefore, it would be nice if you could give some examples (not the full statistical linking between your representativeness statistic and the results from a homogenization method you are referring to) for some stations about a possible link between the values of the representativeness statistics and a break point in the GPS time

15    series. Moreover, in this context, it can also be very instructive to show the time series (probably best the monthly mean time series, although most of the representativeness statistics are calculated from daily values) of both the GPS and ERA-Interim for some example outlying sites (section 4). This can make the discussion of the characteristics of the outlying site differences between GPS and ERA-Interim less heavy to read and might give you the opportunity to illustrate why your method might be an interesting tool to identify the stations for which the GPS-ERA-Interim IWV differences cannot be used for breakpoint

20    identification.

We actually answered the comments about the homogeneity issues but it was clearly stated in the manuscript discussion of the effect of inhomogeneities in either the GPS or the reanalysis data is not the purpose of this paper. Here we want to analyse the cause of systematic model and observation differences that show up in the statistical comparison parameters (mean and standard deviation of differences). We want to emphasize the role of representativeness errors and their variation as a function

25    of space, time, and climatic conditions. We know that the impact of breaks in the GPS or ERAI time series on these statistical parameters is rather small compared to the representativeness errors due to the coarse model resolution. This idea is actually supported by a previous publication by Parracho et al., 2018, and by the strong systematic seasonal variation in all parameters (see Fig. 8 and 9). However, we agree that further insight into the nature of the discrepancies can be brought by inspecting the time series. We inspected time series of daily IWV data (not shown) but the day-to-day variability is usually so large that it is

30    difficult to see the small inhomogeneities. They can be better seen on monthly time series. So we refer now to Figure B2 of Parracho et al., 2018, where monthly time series of IWV and IWV differences are shown for four of the outlying sites discussed in Section 4.

=> sentence added P8: "The time series of IWV and IWV differences for four of the worst cases (CFAG, KIT3, MCM4, and SYOG) can be found in Figure B2 of Parracho et al., 2018."

From this figure it can be seen that CFAG and KIT3 have a strong seasonal modulation in the mean IWV differences. The same information can be seen in our Figure 8a,b for all 15 stations. In addition, our Figure 8c,d show that all 15 sites exhibit a rather strong seasonal modulation either in absolute or relative standard deviation which is not related to breakpoints (breakpoints would not have a systematic effect). So we don't think it is necessary to include more examples of times series.

Regarding the implication of this work to the homogenization topic, we added a sentence P12:

"unless the homogenization method explicitly models the seasonality in the bias and the non-stationarity of the noise variance (Quarello et al., 2018)."

Also the point raised by the second reviewer about "the current knowledge of the spatial representativeness error of the GPS derived PWV and *whether and* how this study tackles this issue" has not been treated in the manuscript. This is however a very important point. Also the first reviewer has some difficulties with interpreting the definition of representativeness error (P2,L32-34: Are representativeness errors never to be referred to model errors? I interpret the definition of a representativeness error as that the only cause is the limited model resolution? If this is correct it can be stated explicitly, because I can also argue that the limited resolution of a model can be the cause of "model errors") and your answer has not been implemented in the manuscript itself. In this context, I miss a reference in your manuscript to earlier papers investigating the concept of representativeness error (the second reviewer already provided a reference to the work by Liou et al. 2001). One such a publication that I missed in particular is the ACP paper by Buehler et al. (2012). In their section 3.5, they give a nice description of the concept of representativeness error, in general terms, and distinguish between two effects: (i) the measurements may not be perfectly collocated in space and time, and (ii), they may have different sampling characteristics – one may be an instantaneous point measurement and the other an average over some distance in space and time. To my opinion, you should be more specific about which representativeness you are considering in the manuscript, which not (or are cancelled out by exact temporal matching e.g.) and how this related to earlier concepts of representativeness. As already mentioned, I also missed references to past studies explaining for which area (cone) a GPS IWV measurement is representative. With this more specific definition of representativeness error, you can also be more specific about the motivation and applications of your study, another remark made by the two reviewers ("applications are not very appealing" and "the stated motivation for the work was to identify GPS stations where ERA Interim is not recommended to be used when searching for inhomogeneities in the GPS time series of IWV").

The Introduction has been significantly revised to address these points:
- We enhanced the discussion of representativeness errors in the data assimilation context and cite 3 three references (Lorenc 1986; Janjic 2006; Waller 2014)
- We discuss the representativeness errors/differences in the context of IWV measurement comparisons and now cite Liou 2001 and Buehler 2012.
- We better define what we call representativeness errors in this study (i.e. the effect of the coarse spatial resolution of the reanalysis)
- We removed the reference to model errors because it is ambiguous and can be understood in different ways (e.g. model errors traditionally refer to errors in model physics/parameterizations, but in the assimilation context they

rather refer to errors in the background forecast) and we cannot actually isolate the contribution of model errors here. Reanalysis data also include errors from the assimilation scheme which in the present context would be included into "model errors".

- The concept of representativeness error discussed by Buehler 2012 and the idea that GPS measurements are representative of a cone and not a straight vertical line are not emphasized here because: (i) the representativeness errors considered in this study are from the model/reanalysis and not from GPS; (ii) the model/reanalysis representativeness errors are typically much larger than observation system sampling/measurement characteristics (Janjic 2006) which is also true here for GPS; (iii) discussing the GPS sampling/measurement characteristics would thus distract the message. We simply take the GPS measurements as the reference.

- We reformulated the sentence relative to the implication of this work for homogenization. It should be understood that we propose a methodology to detect outlying sites. After that, the decision to reject or keep the stations depends on the capability of the homogenization method to account for a seasonality in the bias and noise variance. This is also more explicitly written in the Conclusion section as mentioned above.

We hope these clarifications and corrections make the paper more appealing.

Other examples where you should include your answer to a reviewer comment in the manuscript are:

"RC1: Related to this it is clearly stated that the method is subjective. Nevertheless, if the method is to be applied by others, it would be informative to also document the reasoning behind the choices. For example, why did you choose non-symmetric limits for the mean differences in Figures 2 and 3?"

Done: "The reason why we chose non-symmetric limits with respect to zero for the mean differences is because the distribution is not centred on zero."

" RC1: P4,L9-11: I think you should mention that the GPS time series used have passed some kind of quality check, because a very large break should have an impact on the overall standard deviation of the differences GPS – ERA Interim": be more specific on which data screening has been applied to detect e.g. very large breaks.

It was already mentioned: "ZTD estimates… are first screened for outliers as described in Parracho et al., 2018."

Note that the purpose of the screening is not to detect breaks but outliers.

"RC1: P6,L5: It cannot be taken for granted that the discrepancy is not due to GPS errors just because the formal errors do not increase. For example, a nearby installation of say a metallic structure may introduce significant multipath errors without affecting the formal errors."

We don't agree with this statement. See our answer to the referee's comment.

"RC1: P7,L2-5: Can you compare this standard deviation of 0.81 kg/m2 to what is obtained for stations located in the same area of the present study, in order to quantify the improvement obtained for the higher resolution model?"

Done: "The median standard deviation of GPS-ERAI differences over 12 stations in the same domain amounts to 0.98 kg m-2, so there is clear benefit of higher resolution and more modern physics."

"RC1: P7,L13-15: Perhaps the GPS sites that do not show an improvement using bi-linear interpolation are located close to one of the four grid points that is more representative compared to the others?"

Actually this question is not relevant (and the referee could check it directly from Figure 7), this is why we didn't mention it in the manuscript.

5 We also provide all the numerical results presented in the figures in a supplement for the readers who want to make their own complementary analysis of our results.

Some          other          specific          comments

P6,L13-15: to which representativeness "differences" in the reanalysis humidity field are you referring to here? Please be more

10 specific.

We meant the model representativeness error, but this is in the context of data assimilation. The sentence was changed to:

"This tendency can be explained by larger representativeness errors in the reanalysis humidity field as a function of altitude as also found by Waller et al. (2014) in the Met Office high-resolution UK variable resolution model."

P6,L23-24: "a mistake which has been made in several past studies and led to the erroneous statement that IWV differences

15 increase towards the equator due to the increasing mean IWV": without a proper referencing to those several past studies in which this statement is literally present, this is a very suggestive statement.

We prefer not to cite the papers which made this erroneous statement. So we removed this sentence.

[revised manuscript text omitted]